# A Study on E-Nose System in Terms of the Learning Efficiency and Accuracy of Boosting Approaches

**DOI:** 10.3390/s24010302

**Published:** 2024-01-04

**Authors:** Il-Sik Chang, Sung-Woo Byun, Tae-Beom Lim, Goo-Man Park

**Affiliations:** 1The Graduate School of Nano IT Design Fusion, Seoul National University of S&T, Seoul 01811, Republic of Korea; ischang@daum.net; 2Digital Innovation Support Center, Korea Electronics Technology Institute, Jeonju 54853, Republic of Korea; swbyun@keti.re.kr; 3Intelligent Information Research Division, Korea Electronics Technology Institute, Seongnam 13488, Republic of Korea; tblim@keti.re.kr; 4Department of Smart ICT Convergence Engineering, Seoul National University of S&T, Seoul 01811, Republic of Korea

**Keywords:** electronic nose system, gas recognition, artificial olfactory, environmental application, gas sensor

## Abstract

With the development of the field of e-nose research, the potential for application is increasing in various fields, such as leak measurement, environmental monitoring, and virtual reality. In this study, we characterize electronic nose data as structured data and investigate and analyze the learning efficiency and accuracy of deep learning models that use unstructured data. For this purpose, we use the MOX sensor dataset collected in a wind tunnel, which is one of the most popular public datasets in electronic nose research. Additionally, a gas detection platform was constructed using commercial sensors and embedded boards, and experimental data were collected in a hood environment such as used in chemical experiments. We investigated the accuracy and learning efficiency of deep learning models such as deep learning networks, convolutional neural networks, and long short-term memory, as well as boosting models, which are robust models on structured data, using both public and specially collected datasets. The results showed that the boosting models had a faster and more robust performance than deep learning series models.

## 1. Introduction

Over the past few decades, the field of e-nose research has developed rapidly as part of the general development of IT technologies. The electronic nose is a technology that uses a sensor array designed to mimic the odor detection ability of the human nose to detect and identify odors of specific compounds. In recent years, electronic nose technology has comprehensively overhauled traditional odor detection methods. Traditional odor detection technologies focused on only one odor and were limited in their accuracy and reliability. In contrast, electronic noses provide detection capabilities for a range of chemical substances and have the advantage of being able to collect and process data in real-time. As a result, the potential for application is increasing in various fields, such as medical diagnosis, food quality improvement, gas leak measurement, detection of chemicals such as drugs, and virtual reality [1,2,3,4,5,6,7,8]. However, even with the extensive technological development of electronic noses, there are still various unresolved problems. Improvements are needed in many aspects, including sensor sensitivity, pattern recognition algorithm accuracy, and field applicability. Additionally, the development of electronic nose systems optimized for specific applications requires continued research and effort.

Since Taguchi commercialized the SnO_2_-based MOx (metal oxide) gas sensor in 1968, MOx gas sensors have been widely used due to their various advantages (high sensitivity, low cost, long lifespan, simple peripheral circuit, etc.). MOx sensors are excellent at detecting a wide range of gases and provide relative outputs as gas density changes. In 1982, after the introduction by Persaud and Dodd [9] of the concept of mimicking the human sense of smell using sensor arrays and pattern recognition technology, Gardner and Bartlett [10] defined the electronic nose as a configuration of a sensor array that does not have complete selectivity. Subsequently, the MOx gas sensor was adopted as the main e-nose sensor and its low selectivity was dramatically improved by implementing pattern recognition technology using arrays [11,12]. In recent electronic nose-related research, studies using deep neural networks, such as convolutional neural networks (CNN) and recurrent neural networks (RNN), have been proposed along with the development of artificial intelligence and deep learning technology [13,14,15,16].

Fang et al. observed that because existing electronic nose systems depend on the knowledge of domain experts, their performance may be limited due to information loss during the feature extraction process. To overcome this, the All-Feature Olfactory Algorithm was developed, which makes comprehensive use of adaptive global dynamic information by combining CNN, RNN, and attention technologies [15]. Bilgera et al. proposed a study to estimate the location of gas emission points by processing signals obtained from gas sensors at various locations in a given environment. For this purpose, the location of the gas source was modeled and estimated using the CNN-LSTM (long short-term memory) model [16]. In previous research, the authors of this paper investigated and analyzed models that can minimize the drift effect that may occur when using a MOx sensor, utilizing various deep learning models such as CNN or RNN. However, most of the public datasets used in electronic nose research [17] and the data collected in these studies have been shown through other research to exhibit low levels of data complexity that not only do not require the use of deep models such as deep learning models but also reflect low sensitivity to changes in sensor signals over time. Therefore, it may be more efficient to use statistical or ML approaches rather than the deep learning models utilized in previous research.

In this study, we characterize electronic nose data as structured data and investigate and analyze the learning efficiency and accuracy of deep learning models that use unstructured data. For this purpose, we use the MOx sensor dataset collected in a wind tunnel, which is one of the most popular public datasets in electronic nose research [17]. This dataset has been utilized for gas classification algorithms, gas source location prediction, and other application programs. Additionally, a gas detection platform was constructed using commercial sensors and embedded boards, and experimental data were collected in a hood environment such as used in chemical experiments. We investigate the accuracy and learning efficiency of deep learning models such as DNN, CNN, and LSTM, as well as boosting models, which are robust models on structured data, using both public and specially collected datasets.

The remainder of this paper is organized as follows: Section 2 presents the related works on the e-nose system. Section 3 and Section 4 explain the design of gas detection and experiments, respectively. Section 5 concludes this work.

## 2. Related Work

Since the term “electronic nose” appeared in the late 1990s to describe sensor arrays that could distinguish between one or more chemical components, various approaches have been proposed in research to address the problem of the e-nose. This section describes studies on the conventional e-nose and recent research applying deep learning models.

### 2.1. Conventional E-Nose Studies

Conventional e-nose approaches based on gas detection have relied on traditional data analysis techniques such as principal component analysis (PCA), cluster analysis, and computational fluid dynamics [18,19,20,21]. In addition, machine learning technology and traditional pattern recognition methods have been used with electronic noses for over 30 years. Many studies have been proposed based on these techniques to identify gases and detect gas leaks. Using various multivariate analysis approaches and PCA, Ref. [22] developed an e-nose system based on multiple sensors that is capable of detecting and identifying three explosives. Using both simulated and real data, an artificial neural network (ANN) was utilized with e-nose technology to detect gas leaks at test locations [23]. In another study [24], a feature selection approach to identify gases and a hybrid approach based on the fusion of multiple classifiers were proposed, achieving gas type recognition and concentration levels of 99.73% and 97.54%, respectively. An e-nose system based on six MOx sensors was proposed by Zhang et al. The authors detected flammable and hazardous gases, extracting not only frequency characteristics but also time information [25]. Another e-nose system based on three MOx sensors was proposed by Manjula et al. In order to detect the gas present in the air, the authors utilized time signals as features, submitting them to five different machine-learning classifiers. Among these, the Random Forest (RF) classifier achieved the highest accuracy, reaching an impressive 97.7% [26]. Similarly, to identify gases in the atmosphere, Ragila et al. [27] used six MOx sensors; the accuracy of their ANN using time signals as an input was 93.33%.

### 2.2. Deep Learning-Based E-Nose Studies

Research on the deep learning-based e-nose has developed rapidly in recent years. The electronic nose is used to distinguish certain odors of chemical materials, and deep learning technology contributes to improving and automating them. Sharabiani et al. investigated the efficacy of electronic noses and other chemometric methods, such as principal component analysis, linear discriminant analysis, and the Artificial Neural Network, as cost-effective, rapid, and non-destructive methods for the detection of pure and adulterated rice varieties [28]. Viciano-Tudela et al. proposed the use of gas sensors to detect the adulteration process in the essential oil of Cistus ladanifer and compared the suitability of the tested sensors for detecting adulterated oil and the required measuring time. Gas sensors were used in a measuring chamber to measure pure and adulterated oils [29]. In Ref. [30], the authors used an array of eight different gas sensors to determine gas concentrations. In this task, CNN was used to perform gas classification. In Ref. [31], a framework for pipeline gas leak mitigation was proposed. The authors achieved 92% accuracy by performing the detection process using multiple deep learning (DL) models, including CNN, LSTM, and autoencoders. Pan et al. adopted the deep learning approach using a hybrid framework of CNN and LSTM. Similarly, Ref. [32] used CNN and LSTM to obtain spacetime information to detect gas leaks using limited simulation data. It has been shown that the DL system can learn the characteristics of the value of the gas sensor and classify the data more accurately. The hybrid Deep Belief Network and high-speed gas identification technology based on stacked automatic encoders were presented by Ref. [33]. All previous methods were based on data from gas sensors using sequential procedures. However, as mentioned earlier, relying solely on gas sensor-based detection and deep learning-based identification technologies presents some challenges.

## 3. Design and Development of the Gas Detection Platform

Electronic devices and data analysis procedures must be specifically tailored to develop an electronic olfactory system. This chapter explains hardware and software design for collecting reliable data.

### 3.1. Gas Detection Platform Hardware Design

In general, gas detection as an electric signal has been found to be stable in various environments and temperature ranges and to respond sensitively and quickly to various concentrations and potential analytes. To meet these conditions, MOx sensors have frequently been used in research and have shown good performance. In this study, following these approaches, a gas detection platform was constructed based on the gas detection platform design used in previous e-nose research [17]. The gas detection platform detects a variety of analytes in a demonstration facility and tracks changes in the generated chemical analyte gas. The gas detection platform’s gas sensor measures changes in electrical conduction, captured in time series through the MOx film via interactions that reduce or oxidize the analyte gas on the film’s surface. A multivariate response is displayed to various odoriferous or odorless gas stimuli discharged throughout the sensor electrodes. For this reason, in most studies, multiple sensors rather than a single sensor are used in the gas detection platform, with the sensor configuration varying depending on the type of gas to be detected. In our research, when a small amount of gas was used for an empirical experiment, gases that were relatively harmless to the human body were selected based on the advice of a chemical expert; the type, characteristics, and precautions associated with the selected gas are shown in Table 1. The selected gases mentioned in Table 1 were used in the experiments. We used the gases by dropping two drops (200 µL) using a micropipette into the Petri dish without measuring the concentration of the gases.

Arduino, an open-source computing platform and software development environment based on a microcontroller board, is associated with a digital device that can process commands and controls by connecting interactive objects such as sensors or components. To construct a gas detection platform, this research used the Arduino Nano board, a small microcontroller (ATmega328P) board developed by Arduino.cc with a compact form factor that has 32 KB of memory consisting of 30 pins and a clock speed of 16 MHz, consisting of eight ADC ports that can receive analog sensor values [34]. As mentioned above, there are various types of sensors depending on the gas to be detected; these sensor types are used to find and classify various gases. The Arduino sensor, known as the MQ gas sensor, detects a variety of gases, including alcohol, smoke methane, LPG, hydrogen, ammonia, benzene, and propane [35]. A total of four sensors (MQ-3, MQ-135, MQ-137, and MQ-138) were used in this research, as follows. First, the MQ-3 gas sensor module can detect carbon monoxide, methane, benzene, hexane, LPG, and alcohol, and features high sensitivity and fast response speed; the sensitivity of the sensor can be changed using a potentiometer. This gas sensor module is highly sensitive to alcohol and can also be used as a breathalyzer. Second, the MQ-135 gas sensor module can identify hazardous gases and smoke, including ammonia, sulfur, benzene, and carbon dioxide. Like other MQ Series gas sensors, this sensor has pins for digital and analog outputs; the digital pin output increases when the amount of gas exceeds a predetermined threshold, which can be adjusted using a potentiometer. The analog voltage generated by the analog output pin can be used to approximately measure the concentration of various gases in the atmosphere. Third, the MQ-137 gas sensor is a sensor for measuring ammonia; using a simple circuit, changes in conductivity can be converted into an output signal of the corresponding gas concentration. This sensor is very sensitive to gaseous ammonia and has excellent detection performance for organic amines such as trimethylamine and calamine. Lastly, the MQ-138 gas sensor can detect benzene, toluene, alcohol, acetone, propane, formaldehyde, and hydrogen gas, and is particularly sensitive to toluene, acetone, alcohol, and methanol. It can also be used to monitor hydrogen and organic vapors. The configuration and design of the sensor and board used in this research are shown in Figure 1.

All four sensors operate at 5 V, and the pin configurations consist of VCC, GND, DO (Digital Out), and AO (Analog Out). In this research, VCC, GND, and AO are used, with AO connected to the ADC pin of the Arduino board. Arduino Nano has a 10-bit resolution providing sensor values ranging from 0 to 1023. In order to collect all data from each sensor board, the half-duplex RS-485 communication method is used as shown in Figure 1. RS-485 is a communication method capable of a transmission distance of approximately 1.2 km at a speed of 100 kbps. Using a PC or Raspberry Pi as the master, the slave’s sensor data are collected at regular intervals by initially assigning a different ID to each sensor board.

### 3.2. Gas Detection Platform Software Design

Boosting is a type of ensemble learning, a machine learning technique that combines multiple weak learners to create a strong learner. Boosting algorithms work by training models sequentially and compensating for errors in previous models. Well-known boosting algorithms include AdaBoost, gradient boosting, XGBoost, and LightGBM.

Structured data refers to tabular data consisting of rows and columns, where each row generally represents an observation and each column represents a feature. Boosting algorithms are among the algorithms primarily suited to structured data. The key aspect of the relationship between boosting algorithms and structured data is that boosting algorithms show high performance on structured data because they combine weak learners to generate a strong prediction model. Boosting trains the next model by focusing on the errors of the previous model. Structured data can contain various types of features, such as categorical and numeric features, and boosting algorithms are well-suited to handling various feature types and capturing the interactions between features. For this reason, boosting algorithms show excellent performance on structured data, handling a variety of characteristics well, and are effective in reducing overfitting. Therefore, a boosting algorithm may be more suitable than a complex deep learning model when creating an e-nose model.

Adaboost, the original boosting algorithm, functions by simply increasing the weight of misclassified data points in the decision tree so that they are better classified on the next pass. The concept of the gradient boost model, on the other hand, is to find the value that minimizes the loss function by applying the gradient descent algorithm rather than increasing the weight of error values. Although the gradient is changed several times by a certain amount in the negative direction of the slope, the major drawback of most boosting algorithms, including this type of gradient boosting model, is the risk of overfitting. To prevent overfitting in gradient boosting, a regularization term is added to the loss function. With this approach, XGBoost is widely known for its excellent performance in regression and classification.

We investigate the accuracy and learning efficiency of deep learning models, such as DNN, CNN, and LSTM, as well as boosting models, which are robust on structured data, using both public and specially collected datasets.

## 4. Experiments

### 4.1. Data Collection and Experiments Using a Chemical Laboratory Hood

This section describes the experimental verification of the proposed gas detection platform. Since gas detection platforms require accurate characterization of chemicals based on data, validation involves tasks ranging from optimizing all parameters at the measurement site to accurate analysis of the chemical reaction. A test bed measuring 2.5 m × 1.2 m × 0.4 m in a hood environment for chemical experiments was used to conduct experiments required by the research and to evaluate the functionality of the gas detection platform. The hood provides an environment in which environmental conditions can be implemented relatively accurately while minimizing disturbance due to external atmospheric flow. Some design details of the hood test bed facility and the datasets and measurement procedures are described in the subsections. An example of hood usage and sensor installation is shown in Figure 2.

Gas detection platforms numbered from 1 to 20 were configured so that the expected airflow in the hood could be measured in the inlet/outlet; to exclude conditions for convective diffusion and to minimize the influence of obstacles, the gas detection platforms were installed, as shown in Figure 3.

In order to acquire high-quality gas data, the following gas collection process was set up. Before injecting and detecting gas, the hood door was opened, the hood wind speed was set to at least 10 m/s, and the hood was ventilated for 1 min. To prepare for gas data collection, the hood door was closed and the wind speed was set to about 1 m/s. The opening and closing of the hood door are shown in Figure 4.

For gas sensor data collection, data were collected for a total of 60 s, and temperature and humidity were also measured over this time. The temperature and range of humidity levels when measuring the gas are about 26 degrees Celsius and 50–53%, respectively. In this study, we did not consider controlling temperature and humidity. The target gases were acetone, ammonia, benzene, ethanol, and toluene. A crucial flaw of MOx sensors is their sensitivity to sensor drift, which comprises unpredictable alterations in the signal response upon continuous exposure to uniform material. Sensor drift is primarily caused by the chemical and physical interactions of a sensor site, such as sensor aging (restructuring of a sensor’s surface over time) and sensor poisoning (irreversible or slowly reversible combinations of previously measured gases or other contaminants). Environmental factors such as changes in humidity, temperature, and pressure also affect sensor response. The effects of sensor drift can be reduced in the experiment planning process by assigning random materials to be exposed in the data selection process. In order to minimize the effect of drift, susceptibility to which is the biggest drawback of the MOx sensor, the procedure was repeated five times per gas before the gas was changed; the gas collection order was ethanol, toluene, acetone, benzene, and, lastly, ammonia. After each set (ethanol to ammonia) was collected, the hood environment was ventilated for 5 min. Ten seconds after starting gas data collection, the target gas was injected into the hood, and 10 s after that (starting at 20 s), the gas was removed from the hood. During the remaining 40 s, the residual gas remaining in the hood was measured. Ventilation was then performed for at least 3 min while the gas was changed.

As a result, in the chemical laboratory hood experiment, five classes (acetone, ammonia, benzene, ethanol, toluene) of sensor data were collected for 60 s and stored in CSV format; the saved signal data of sensors during exposures is shown in Figure 5 below.

### 4.2. Experimental Results on Collected Data

This section describes the comparison test between boosting approaches and deep learning approaches in terms of learning efficiency and accuracy. Spandonidis et al. proposed multiple deep learning models for detecting pipeline gas leaks, including CNN, LSTM, and autoencoders [31]. Chang et al. introduced various gas detection models using deep learning approaches from a temporal and spatial perspective at multiple locations [36]. In this experiment, the LSTM model and 1D CNN + LSTM model proposed by the authors were selected as baselines. The collected data were divided into five classes (acetone, ammonia, benzene, ethanol, and toluene) for model learning. The data used for model learning runs from 15 to 25 s (a total of 10 s) after the gas was injected. A total of 80 data samples collected from sensor 1 on board 1 through sensor 4 on board 20 were used as input vectors. To verify the learning accuracy and learning efficiency of each model, the discrimination accuracy and learning time of the model were measured for the five classes. The dataset was divided in an 8:2 ratio, and cross-validation was performed five times. Table 2 shows the performance results.

According to the results, boosting models show higher performance than deep learning series models, and the simplest machine learning models show lower performance than deep learning series models. Among the former, the XGBoost model showed the highest accuracy at 99.6%. Boosting models also show higher efficiency in inference time and training time.

### 4.3. Experimental Results on Published Data

The published gas sensor dataset comprises data collected from 18,000 time-series signals over 16 months by releasing 10 analysis gases (acetone, acetaldehyde, ammonia, butanol, methane, methanol, carbon monoxide, benzene, and toluene) [17]. Combinations of parameter gas, position, wind speed, and operating voltage were selected prior to measurement until each combination was repeated 20 times. This continued for 260 s, and the gas was released from t = 20 s to t = 200 s. Data were saved in one file for each parameter combination (e.g., CO_1000, Ethylene_500, Acetaldehyde_500, and others).

In the data preprocessing process, we assumed that all data at the midpoint of the time series sensor data were collected after sufficient gas had been injected; the data were sampled as shown in Figure 6 to analyze the signal value when sufficient gas had been injected. 

Data were randomly sampled 10 times from the extracted signal in a specific range. That is, Nx is calculated as (9 × Nd × 10) × 8, where Nd is the number of central data files. Among the gas data, CO_1000 and CO_4000 differ only in ppm; therefore, CO_4000 was excluded from the analysis. The discriminative accuracy of the model for six classes was measured; the dataset was divided in the ratio 9:1, and five-fold cross-validation was performed, as shown in Table 3.

According to the results, and similarly to the results in Section 4.1, boosting series algorithms show higher performance than deep learning series models. Additionally, boosting models show higher efficiency in inference time and training time.

## 5. Conclusions

In this study, we characterized electronic nose data as structured data and investigated and analyzed the learning efficiency and accuracy of deep learning models that use unstructured data. For this purpose, we used the MOx sensor dataset collected in a wind tunnel, which is one of the most popular public datasets in electronic nose research. Additionally, a gas detection platform was constructed using commercial sensors and embedded boards, and experimental data were collected in a hood environment such as used in chemical experiments. We investigated the accuracy and learning efficiency of deep learning models, such as DNN, CNN, and LSTM, as well as boosting models, which are robust models on structured data, using both public and specially collected datasets. According to the results, boosting models showed higher performance than deep learning series models. Among the models, the XGBoost model showed the highest accuracy. Boosting models also showed higher efficiency in inference time and training time.

Future works will require technical research on low-concentration gas detection algorithms, as well as the development of gas detection platforms that can be detected in low-concentration gas. Additionally, research on verifying the capability to discriminate gases also inside mixtures will be required.

## Figures and Tables

**Figure 1 sensors-24-00302-f001:**
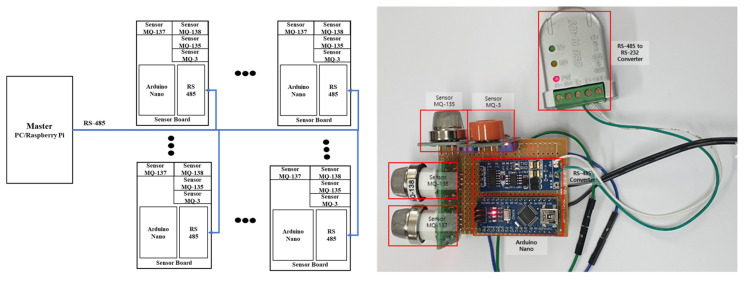
The configuration and design of the sensor and board.

**Figure 2 sensors-24-00302-f002:**
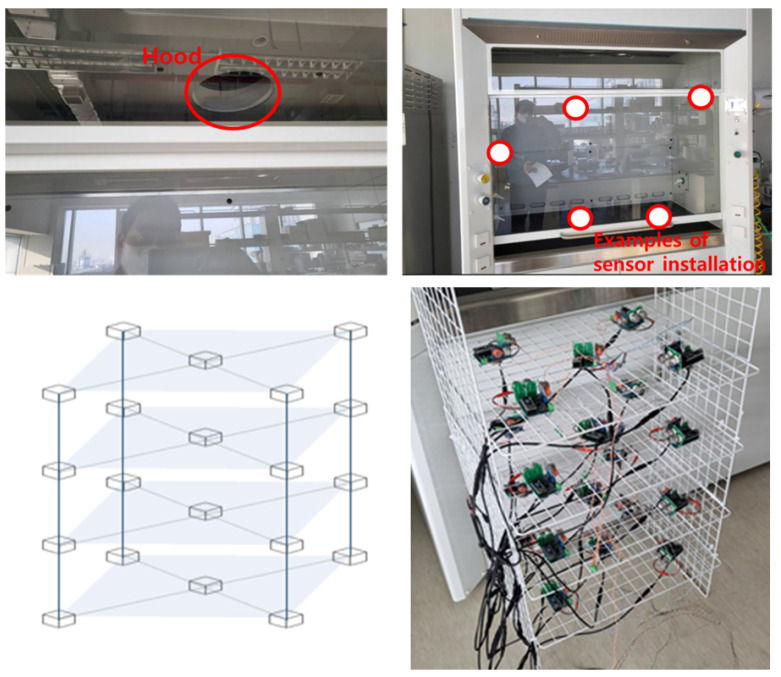
An example of hood usage and sensor installation.

**Figure 3 sensors-24-00302-f003:**
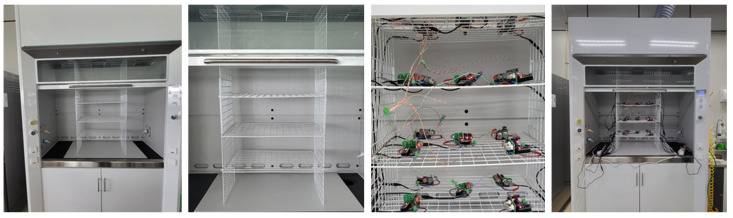
Data collection environment for verification in a chemical laboratory hood.

**Figure 4 sensors-24-00302-f004:**
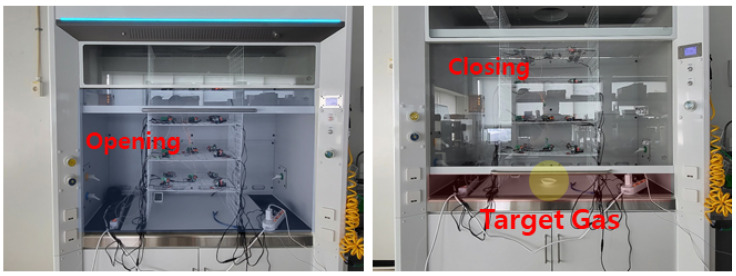
An example of the opening and closing of the hood door.

**Figure 5 sensors-24-00302-f005:**
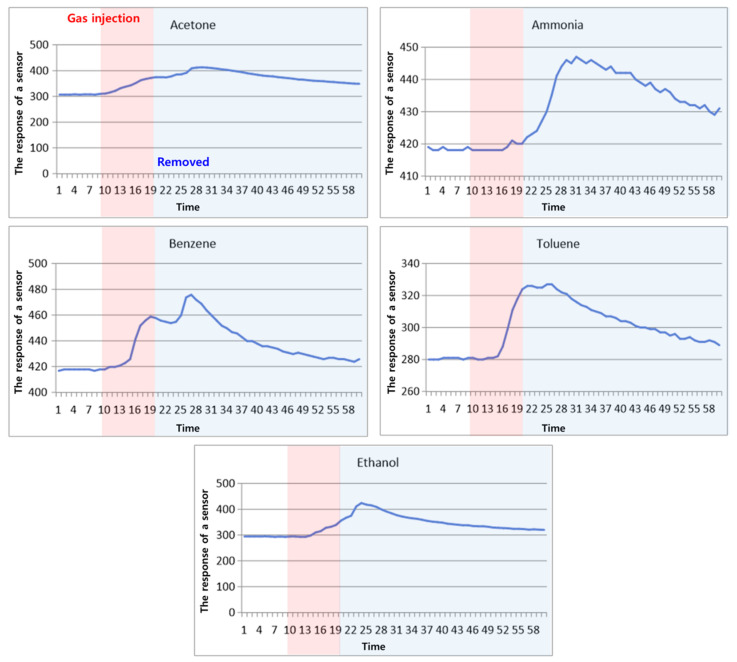
An example of sensor signal during exposures.

**Figure 6 sensors-24-00302-f006:**
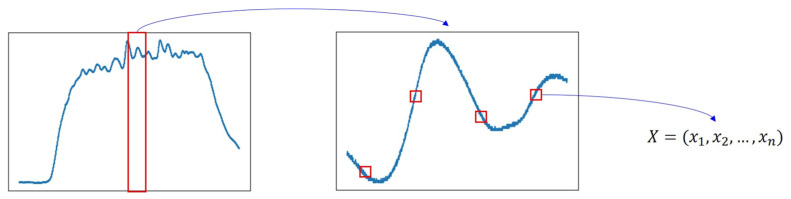
An example of data sampling.

**Table 1 sensors-24-00302-t001:** The type, characteristics, and precautions associated with the selected gas.

Name	Advice	Other
Benzene	Inhaling benzene can lead to drowsiness, dizziness, and loss of consciousness, while prolonged exposure to benzene can affect the bone marrow, leading to anemia and leukemia.	It is recommended to use just one or two drops in the Petri dish.
Toluene	Excessive inhalation may result in functional disorders such as abdominal pain and vomiting, or neurological disorders such as headaches and hallucinations.	
Ammonia	When exposed to high levels of ammonia, it is rapidly absorbed by the mucous membranes, destroying cellular tissue to a lethal level.	It is recommended to use ammonium hydroxide, which is diluted with water.
Acetone	It can be harmful if inhaled into the airway, and long or repeated exposure can damage blood in the body.	Relatively safe.
Ethanol	Ethanol is used to disinfect non-human medical equipment, such as hand disinfection, without wounds.	Relatively safe.

**Table 2 sensors-24-00302-t002:** The comparison results of machine learning models using chemical laboratory hood data.

Model	Accuracy	Training Time
Bayesian	91.8%	1 s
SVM	95.3%	4 s
Linear Regression	95.1%	1 s
Fully Connected Model	97.2%	About 5 min (3000 epochs)
LSTM [31]	96.3%	About 7 min (3000 epochs)
Bidirectional LSTM [36]	97.2%	About 8 min (3000 epochs)
1D CNN + LSTM [36]	98.4%	About 20 min (10,000 epochs)
Temporal CNN [36]	91.2%	About 20 min (10,000 epochs)
AdaBoost	98.1%	0.14 s
Gradient Boosting	99.2%	3.44 s
XGBoost	99.6%	0.23 s

**Table 3 sensors-24-00302-t003:** The comparison results of machine learning models using published data.

Model	Accuracy	Training Time
Bayesian	85%	1 s
SVM	97%	4 s
Linear Regression	85%	1 s
Fully Connected Model	87%	About 10 s
LSTM [31]	89%	About 30 s
Bidirectional LSTM [36]	96%	About 1 min
1D CNN + LSTM [36]	99%	About 5 min
Temporal CNN [36]	98.2%	About 5 min
AdaBoost	96%	0.3 s
Gradient Boosting	99%	5 s
XGBoost	99%	0.4 s

## Data Availability

Data is contained within the article.

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
