# Peer review of "A Study on E-Nose System in Terms of the Learning Efficiency and Accuracy of Boosting Approaches"

_sensors, 2024, doi:10.3390/s24010302_

Round 1
Reviewer 1 Report
Comments and Suggestions for Authors
Reviewer Comments on [sensors-2758108]
Having reviewed the paper titled "[ A Study on e-nose system in terms of the learning efficiency and accuracy of ensemble machine learning approach]" with a focus on its contributions to the field of electronic nose systems and machine learning. While the topic is of significant interest and relevance, I have several concerns that lead me to recommend rejection in its current form. My assessment is as follows:
Minors:
1. Clarity of Figures and a lack of information in the graphical representations
Figures 1, 2, 3, 4, and 5 are not clear. Figure 5 lacks proper labeling of variables and units, which is not in accordance with standard scientific presentation norms.
2. Title and Content Consistency
The manuscript's title implies a comprehensive study of ensemble machine learning methods. However, the content predominantly focuses on the Boosting method. This discrepancy between the title and the content can be misleading for readers.
3. Incomplete Literature References
The paper does not provide adequate citations. For instance, the introduction's statements about the field's expanding applications in medicine, environmental monitoring, and virtual reality, and traditional odor detection technologies lack supporting references. Also, when discussing previous studies using CNN or RNN models to minimize MOX sensor drift effects, relevant research citations are necessary.
4. Contradictions in Argumentation
The manuscript exhibits contradictions in viewpoints, especially in discussions regarding the efficacy and overfitting risks of Boosting algorithms (refer to line 201 and line 214). This inconsistency in argumentation impacts the logical coherence of the manuscript.
Majors:
The theoretical part and experimental design of the paper do not robustly support the thesis. The central arguments of the paper lack persuasiveness.
1. Theoretical Aspects
The key arguments lack sufficient literature support, particularly in the introduction discussing the complexity of electronic nose data and the applicability of deep learning models. This absence weakens the academic foundation and credibility of the arguments.
2. Experimental Design
The selection of deep learning models is very limited and lacks diversity. A narrow focus on a few models may not be representative of the broader applications in deep learning, potentially leading to selection bias and limited applicability and generalizability of the findings. Moreover, the specific parameter settings of the models, crucial in deep learning research, are not detailed, affecting the transparency and reproducibility of the experiments.
Reviewer Recommendations:
1. Consider revising the manuscript title or expanding the research scope to align with the title.
2. Include comprehensive literature citations to substantiate key statements and arguments.
3. Address contradictions in argumentation to enhance logical coherence.
4. Expand the range of deep learning models tested and provide detailed parameter settings for each model.
5. Figures should be revised to include full information, such as units and clear labeling. All graphs need to be clear and accurately labeled to meet scientific standards.
Comments on the Quality of English LanguageModerate editing of English language required
Author Response
Thank you for inviting us to submit a revised draft of our manuscript entitled, “A Study on e-nose system in terms of the learning efficiency and accuracy of ensemble machine learning approach” to Sensors. We also appreciate the time and effort you and each of the reviewers have dedicated to providing insightful feedback on ways to strengthen our paper. Thus, it is with great pleasure that we resubmit our article for further consideration. We have incorporated changes that reflect the detailed suggestions you have graciously provided. We also hope that our edits and the responses we provide below satisfactorily address all the issues and concerns you and the reviewers have noted.
To facilitate your review of our revisions, the following is a point-by-point response to the questions and comments delivered in your letter dated 14-December-2023.
Best regards,

Reviewer 2 Report
Comments and Suggestions for Authors
The authors have presented the use of deep learning models to identify the type of gas using MQ-based sensors. The topic is interesting and aligned with the journal's scope. The major problem of the manuscript is the limited results. The authors have to improve the paper in order to reach the expected impact for its publication in this journal. Following, I include a series of comments aimed at enhancing the quality of the manuscript.
1. The introduction of the analysed problem, lines 10 to 14, in the abstract is too long. Please reduce the provided information to just one or two sentences.
2. Check the double space in line 20.
3. Consider avoiding the use of acronyms in the abstract, especially for those acronyms used once or twice. If the acronym is necessary, define it the first time it is used. Check IT, MOX, DNN, CNN, and LSTM.
4. In the abstract, the authors have to highlight their results, including numerical values of the obtained accuracies with each one of the datasets and the learning models..
5. Add more keywords to have a minimum of 6 keywords.
6. In the introduction and beyond, when an acronym is used, it must be described using the full name and the first letters of each word should be capitalised. An example is metal oxide (MOx).
7. Avoid including old references such as references 1 and 2 unless they are vital for the study. It is suggested to cite a recent survey to contextualise the changes over the last decades.
8. There is a lack of references in the introduction. Please add at least 10 more references in the introduction.
9. Check the double space in line 84.
10. Add the structure of the paper at the end of the introduction.
11. Section 2 should be Related Work, not Related works (no final s).
12. At the beginning of the related work the auhtors have to add a short paragraph introducing the content.
13. In the related work, some recent uses of gas sensors combined with machine learning, such as for adulteration detection, are not included. Please add the following references to their recent use:
A. (2023). Non-destructive test to detect adulteration of rice using gas sensors coupled with chemometrics methods. International Agrophysics, 37(3), 235-244.
B. (2023). Proposal of a Gas Sensor-Based Device for Detecting Adulteration in Essential Oil of Cistus ladanifer. Sustainability, 15(4), 3357.
14. At the beginning of a section which includes several subsections, please add a short paragraph detailing the content of the different subsections. Check Section 3 and other cases.
15. Check the letter size and font in Table 1 and correct according to the template.
16. Clarify in lines 142-145 if all gases included in Table 1 will be used in the tests.
17. It is necessary to add the reference (datasheet or other) of the used commercial devices, such as the Arduino ATMega and sensors. In Table 2, it is possible to add the references for the gas sensors.
18. In Table 2, clarify which of the mentioned sensors will be included in the experiments. A column can be added for this prupose.
19. In section 3, more details about data gathering must be provided. For example, the dimensions in which the gas is released, if there was ventilation or not. Moreover, the details about data gathering, such as the number of nodes, the periodicity of data gathering and the disposal of nodes, must be included.
20. Besides the previous comment, the details about the machine learning models that are used should be provided. Why these models are selected? Which software has been sued for running them? What is the size of the generated dataset? Which validation is used?
21. The section 4 must be divided. Part of this content should be included in the previous section following the previous recommendations. The results should be included in a section named Results.
22. More information on obtained results must be provided to extend the section Results.
23. The authors can compare their results using other simplest machine learning models and discuss about the cost of precision and their impact on energy use due to the deep learning models, which is a current issue.
24. Future work should be added in an independent paragraph at the end of the conclusions.
Author Response

(The authors gave the same response as above.)

Reviewer 3 Report
Comments and Suggestions for Authors
The work presented by Il-Sik et al. reports on different machine learning techniques (including DNN, CNN, LSTM, and boosting models) that can be exploited to analyse data from a sensor array. In particular, they develop an array using commercial MOX sensors and demonstrated that considering the tested analytes (ammonia, toluene, acetone, benzene and ethanol), boosting methods are the ones that give the higher accuracy. The manuscript could be considered promising, but it lack clarity, easiness to follow and proper discussion of the results and of the exploited methods. Unfortunately, I do not think that it could be published in the present form, but substantial revisions are required.
In details:
1) Introduction should be rewritten, since currently is not well focused and some sentences and concepts are repeated few times (for instance: lines 29-31 express the same content of lines 35-38). Additionally, some recent literature on the application of electronic noses should be added (for instance authors should cite: https://doi.org/10.3390/s22020577; https://doi.org/10.3390/nano12172992; https://doi.org/10.1016/j.foodres.2022.112214 )
2) Table 1: information reported there seem quite out of topic. It could be more useful to add the concentration used during the exposures.
3) More dynamical curves (i.e. signal of sensors during exposures) should be reported for all the tested gases.
4) More information on the exposures should be added; for instance, which are the tested concentration of all analytes? What are temperature and humidity values during the exposures? How do the authors deal with humidity?
5) Figure 5: axes labels are missing, please correct.
6) Important parameters in the field of gas sensor, such as recovery time, stability over time and detection limit, are not reported and they should be added and discussed.
7) Authors should add exposures to the mixtures of the tested gases, in order to verify the capability to discriminate gases also inside mixtures.
8) Results section is not clear; the authors should rewrite the paragraph. For instance, what do the authors mean with “Experiments on published data” (line 276)? Where did the data come from? There is no reference at all. Additionally, referring to the sentences “CO_1000 and CO_4000 differ only in ppm; therefore CO_4000 was excluded from the analysis.”, CO_1000 and CO_4000 have never been mentioned before. What are they? Sensors? Why CO_4000 has been excluded? Authors should explain in detail the dataset and their choice of analysis, as well as support the data with a more in depth discussion of the results.
9) Carefully check English and grammar; sentences are not always correct.
Comments on the Quality of English LanguageCarefully check English and grammar; sentences are not always correct.
Author Response

(The authors gave the same response as above.)

Round 2
Reviewer 1 Report
Comments and Suggestions for Authors
The paper can be accepted.
Author Response
We appreciate the time and effort you and each of the reviewers have dedicated to providing insightful feedback on ways to strengthen our paper.
Reviewer 2 Report
Comments and Suggestions for Authors
The comments were correctly addressed
Author Response

(The authors gave the same response as above.)

Reviewer 3 Report
Comments and Suggestions for Authors
Il-Sik et al. revised the manuscript taking into account some of the comments previously made by myself and partially improving quality and clarity of the manuscript. Before publication, I think that the manuscript still requires some revision and all the comments previously made should be properly addressed.
In details:
1) Page 2, sentence in lines 41-43 is exactly the same as in lines 46-49. Please delete one of the two sentences.
2) Page 3, line 26: “Sandra et al.” should be replaced by “Viciano-Tudela et al.”, since Sandra is the personal name.
3) Previous comments 2 and 4 have not been addressed. Which are the concentrations of the analytes used in the present work? This information is quite important and should be reported clearly in the manuscript for all the tested gases.
4) Previous comment 5 remains valid also for new figure 5. Indeed, new figure 5 still lacks proper labelling on the axis, which could not be accepted in a scientific manuscript. Please, add the labels and not simply delate the numbers as done for former figure 5, now figure 6.
5) Additional comment on Figure 5: authors should clearly indicate in the graphs where the exposures start and finish. Indeed, they wrote in the text that the exposure lasts 60 seconds, but for instance, considering the Acetone panel, the increase of the signal (supposedly the exposure) is between 10 (seconds?) and 28, so it lasts about 18 (seconds?). The same occurs for all the other gases. Authors should clearly and properly comment the data.
6) Previous comment 6 has not been addressed; indeed, nor at page 7, line 18 neither at page8, line 2, the reviewer could find information on the detection limit of the sensors, as the authors wrote in the reply. Additionally, authors should comment also on stability and recovery time, as previously asked by the reviewer.
Author Response
Concern # 1: Page 2, sentence in lines 41-43 is exactly the same as in lines 46-49. Please delete one of the two sentences.
Author response: According to reviewer’s comment, we have removed the sentence in 2.1 section
We updated the manuscript by (2.1 section).
Concern # 2: Page 3, line 26: “Sandra et al.” should be replaced by “Viciano-Tudela et al.”, since Sandra is the personal name.
Author response: We have modified the issue.
We updated the manuscript by (page 3, line 22).
Concern # 3: Previous comments 2 and 4 have not been addressed. Which are the concentrations of the analytes used in the present work? This information is quite important and should be reported clearly in the manuscript for all the tested gases.
Author response: When we conducted the experiment, we measured the natural diffusion of the gases through sensors by dropping two drops into the petrischale without measuring the concentration of the gas. In order to clarify this part, we have added this information above Table 1.
Temperature and humidity did not change significantly during the measurement of the gases, so we have added a range of temperature and humidity during the measurement.
We updated the manuscript by (page 4, lines 12-13 and page 7, lines 11-12).
Concern # 4: Previous comment 5 remains valid also for new figure 5. Indeed, new figure 5 still lacks proper labelling on the axis, which could not be accepted in a scientific manuscript. Please, add the labels and not simply delate the numbers as done for former figure 5, now figure 6.
Author response: We have added the labels on the axis.
We updated the manuscript by (figure 5).
Concern # 5: Additional comment on Figure 5: authors should clearly indicate in the graphs where the exposures start and finish. Indeed, they wrote in the text that the exposure lasts 60 seconds, but for instance, considering the Acetone panel, the increase of the signal (supposedly the exposure) is between 10 (seconds?) and 28, so it lasts about 18 (seconds?). The same occurs for all the other gases. Authors should clearly and properly comment the data.
Author response: We already mentioned
"For gas sensor data collection, data were collected for a total of 60 seconds." and
"Ten seconds after starting gas data collection, the target gas was injected into the hood, and 10 seconds after that (starting at 20 seconds), the gas was removed from the hood. During the remaining 40 seconds, the residual gas remaining in the hood was measured.".
In order to clarify this, we have modified the figure 5.
We updated the manuscript by (figure 5).
Concern # 6: Previous comment 6 has not been addressed; indeed, nor at page 7, line 18 neither at page8, line 2, the reviewer could find information on the detection limit of the sensors, as the authors wrote in the reply. Additionally, authors should comment also on stability and recovery time, as previously asked by the reviewer.
Author response: We think we already mentioned the limit of the sensors and stability, and we set up the process of data collection, considering recovery time. (page. 7, lines 14-26)
A crucial flaw of MOx sensors is their sensitivity to sensor drift, which is unpredictable alterations in the signal response upon continuous exposure to uniform material. Sensor drift is primarily caused by chemical and physical interactions of a sensor site, such as sensor aging (restructuring of a sensor's surface over time) and sensor poisoning (irre-versible or slowly reversible combinations of previously measured gases or other contam-inants). Environmental factors such as changes in humidity, temperature, and pressure also affect sensor response. The effects of sensor drift can be reduced in the experiment planning process by assigning random materials to be exposed in the data-selection process. In order to minimize the effect of drift, susceptibility to which is the biggest drawback of the MOx sensor, the procedure was repeated five times per gas before the gas was changed; the gas collection order was ethanol, toluene, acetone, benzene, and lastly ammonia. After each set (ethanol to ammonia) was collected, the hood environment was ventilated for 5 minutes.
If this is not enough, please let us know where the information should be written and we will definitely reflect it.